# Regio- and enantioselective CuH-catalyzed 1,2- and 1,4-hydrosilylation of 1,3-enynes

Zi-Lu Wang[1], Qi Li[1], Meng-Wei Yang[1], Zhao-Xin Song[1], Zhen-Yu Xiao[1], Wei-Wei Ma[1], Jin-Bo Zhao[2] ✉ & Yun-He Xu ®[1] ✉

We report a copper-catalyzed ligand-controlled selective 1,2- and 1,4-hydrosilylation of 1,3-enynes, which furnishes enantiomerically enriched propargyl- and 1,2-allenylsilane products in high yields with excellent enantioselectivities (up to 99% ee). This reaction proceeds under mild conditions, shows broad substrate scope for both 1,3-enynes and trihydrosilanes, and displays excellent regioselectivities. Mechanistic studies based on deuterium-labeling reactions and density functional theory (DFT) calculations suggest that allenylcopper is the dominant reactive intermediate under both 1,2- and 1,4-hydrosilylation conditions, and it undergoes metathesis with silanes via selective four-membered or six-membered transition state, depending on the nature of the ligand. The weak interactions between the ligands and the reacting partners are found to be the key controlling factor for the observed regioselectivity switch. The origin of high enantiocontrol in the 1,4-hydrosilylation is also revealed by high level DLPNO-CCSD(T) calculations.

Organosilane compounds are widely used in organic synthesis and material science[1–3]. Among them, propargylsilanes[1,2,4–6] and allenylsilanes[7,8] are versatile synthetic building blocks in the synthesis of pharmaceuticals and natural products. Various methods have been reported for the synthesis of allenylsilanes, and most of them are based on the $S_N2'$ substitution reactions of propargyl alcohol derivatives[9–18]. Traditional methods for the synthesis of propargylsilane usually require harsh reaction conditions and/or the use of highly reactive propargylic Grignard reagents or alkynyl lithium reagents[19–22]. Only limited examples realized efficient synthesis of propargylsilane under mild conditions[23–26].

Functionalization of 1,3-enyne, a type of readily available compounds widely used in organic synthesis, has emerged as a powerful method to construct allenyl- and propagyl-derivatives[27–29]. However, hydrosilylation reaction of 1,3-enynes remains underdeveloped, in contrast to the significant advances in the hydrosilylation of alkenes[30–34], alkynes[35–40] and 1,3-dienes[41–50]. Such reactions can proceed through three main pathways, including 1,2-/3,4-[51–57], and 1,4-hydrosilylations[58–62], which makes control of regio-, stereo-, and enantioselectivity difficult (Fig. 1a). As an example, due to the high reactivity of alkyne moiety, 3,4-hydrosilylation of 1,3-enyne is the most

common reaction type[51–57]. To the best of our knowledge, selective 1,2-hydrosilylation reaction of 1,3-enynes has not been described. Owing to the high reactivity of allene moiety, an additional challenge in 1,4-hydrosilylation of 1,3-enynes comes from the undesired further hydrosilylation of allenylsilane product[63–65]. In addition, only a couple of examples realized selective 1,4-hydrosilylation of 1,3-enynes, and most of them exhibited narrow substrate scope[58–62]. Most recently, Cui and co-workers reported an rare-earth-catalyzed selective 1,4-hydrosilylation of 1,3-enynes, but this strategy is only suitable for alkyl-substituted 1,3-enynes (Fig. 1b)[62]. An addition challenge rises from enantiomeric control. Reports on synthesis of optically pure allenylsilanes via hydrosilylation reaction are rare. In 2001, Hayashi group realized the first asymmetric 1,4-hydrosilylation of 1,3-enynes with Pd/(S)-(R)-bisPPFOMe catalytic system, but moderate enantiomeric excess of products were obtained (18–90%) (Fig. 1b)[60]. Moreover, the yield and enantioselectivity of products cannot be efficiently controlled simultaneously. In this context, the development of novel and efficient catalytic system for the convenient synthesis of propargyl- and allenylsilane products is still highly desired.

Regiodivergent reactions provide a promising tool to obtain multiple regioisomers from the same material. The exploitation of new

[1]Department of Chemistry, University of Science and Technology of China, 230026 Hefei, P. R. China. [2]Faculty of Chemistry and Life Science, Changchun University of Technology, 130012 Changchun, P.R. China. ✉e-mail: zhaojinbo@ccut.edu.cn; xyh0709@ustc.edu.cn

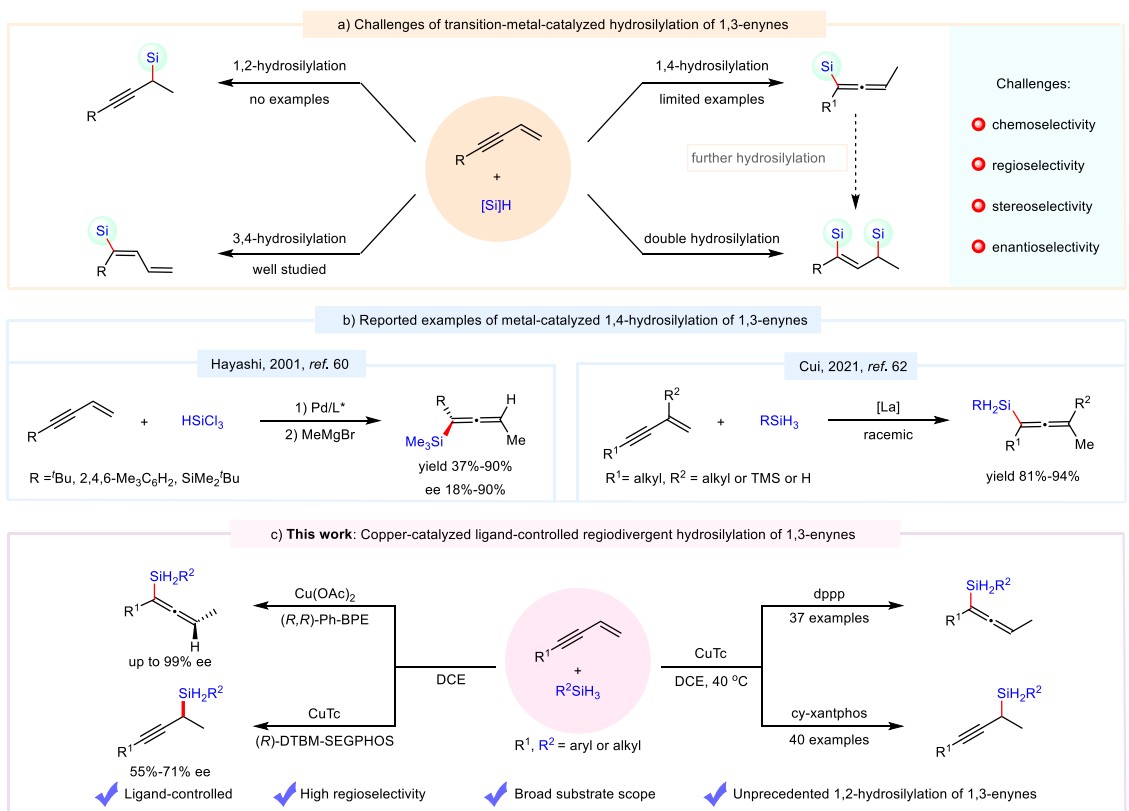

**Fig. 1 | Transition-metal catalyzed hydrosilylation of 1,3-enynes. a** Challenges of transition-metal-catalyzed hydrosilylation of 1,3-enynes. **b** Reported examples of metal-catalyzed 1,4-hydrosilylation of 1,3-enynes. **c** This work: Copper-catalyzed ligand-controlled regiodivergent hydrosilylation of 1,3-enynes.

regiodivergent reactions that deliver multiple regioisomers has become an active field of research over the past several years[66–72]. Encouraged by the excellent performance of copper catalysts in the regioselectivity regulation of the hydrosilylation of 1,3-dienes and allenes[46,48,63], we became interested in exploring copper-catalyzed regiodivergent hydrosilylation reaction of 1,3-enynes toward distinct products. Herein, we communicate the results of copper-catalyzed ligand-controlled regiodivergent 1,2- and 1,4-hydrosilylation of 1,3-enynes (Fig. 1c).

## Results
### Optimization of reaction conditions
We commenced our study by using but-3-en-1-yn-1-ylbenzene (**1a**) and phenylsilane (**2a**) as the model substrates. The optimizations of reaction conditions are shown in Table 1. Initially, different solvents were surveyed with CuTc as the catalyst precursor and dppp as the ligand. Unfortunately, the undesired side product **4aa** was found in moderate yield (42–63%), while 1,4-hydrosilylation product **3aa** was not observed (entries 1-3). We reasoned that highly reactive allene fragment of **3** may react with another silane to form the byproduct **4**. To our delight, when chlorinated solvents such as DCM, DCE, DCP and DCB were used, moderate yields of target product **3aa** were observed accompanied with the formation of double hydrosilylation byproduct **4aa** (entries 4−7). (the effect of chlorinated solvent see Supplementary Fig 1) Subsequently, reducing the loading of **2a** to 3.0 equivalents and lowering the temperature to 40 °C could improve the yield to 64% (entry 8). To further improve the yield of **3aa**, we next examined a range of phosphine ligands, but all resulted in poor yields (entries 9–11). The ratio of CuTc to dppp had an obvious impact on product yield. By increasing the amount of dppp to 20 mol%, the yield of **3aa** was improved to 77% and production of side-product **4aa** was appreciably suppressed (entry 12). Considering the high reactivity of phenylsilane may lead to the formation of byproduct **4**, the less reactive *n*-octylsilane (**2b**) was

used. The corresponding 1,4-hydrosilylation product was obtained in a higher yield (entry 13). Surprisingly, when xantphos was used as ligand, the ratio of 1,2-hydrosilylation product **5aa** was improved obviously (entry 14). We envisioned that the steric hindrance of ligand may play an important role on the control of regioselectivity. Changing the phenyl group of xantphos to the bulkier cyclohexyl group improved the yield of **5aa** to 90% (entry 15). Reducing the loading of **2a** to 1.5 equivalents had no erosion on the yield or regioselectivity (entry 16).

### Substrate scopes
With the optimal reaction conditions in hand, we set out to investigate the substrate scope of 1,3-enynes and silanes (Fig. 2). First, the electronic and steric effects of the 1,3-enynes were evaluated (**3bb**–**3sb**). The substrates bearing alkyl groups or halogen atoms on the phenyl ring delivered the corresponding products in high yields (**3bb-3db, 3gb-3kb**, 75–84%). Besides, different electron-donating or electron-withdrawing groups, such as –OMe, –NPh₂, –NBzMe, –OTs, –OCF₃ –CF₃, –COOMe, could all be well tolerated, and the corresponding products were obtained in good yields (**3lb-3sb**, 69–82%). In addition, the heteroaryl substituted enynes were also proved suitable substrates under the reaction conditions (**3tb-3ub**, 79–80%). It is noted that vinyl or alkynyl groups on the phenyl ring could also be well tolerated (**3vb-3yb**, 64–78%). Less reactive 1,3-enynes **1z** and **1aa** were also competent, affording the corresponding products in moderate yields under modified conditions (**3zb-3aab**, 68–69%). The generality of different silanes was also investigated. Installation of electron-donating or electron-withdrawing group on the *para*- position of phenyl ring led to the desired products in moderate yields (**3ac-3ah, 3ei**, 61–78%). When PhMeSiH₂ was used as silane source, the corresponding product **3aj** was obtained in 55% yield with 64:36 diastereoselectivity. However, the use of more steric hindered Ph₂SiH₂ only produced the desired product in 15% yield. Unfortunately, 1-substituted or

## Table 1 | Copper-catalyzed regiodivergent hydrosilylation of 1,3-enynes[a]

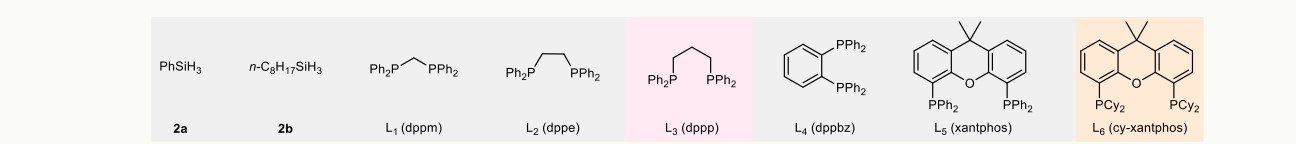

| Entry | RSiH₃ (equiv) | Ligand (mol %) | Solvent (1.0 mL) | T/ °C | t/h | yield % | | |
|---|---|---|---|---|---|---|---|---|
| | | | | | | **3** | **4** | **5** |
| 1 | **2a** (4.0) | L₃ (10) | Cy-H | 60 | 1.5 | 0 | 42 | 1 |
| 2 | **2a** (4.0) | L₃ (10) | MTBE | 60 | 1.5 | 0 | 58 | 1 |
| 3 | **2a** (4.0) | L₃ (10) | THF | 60 | 1.5 | 0 | 63 | 1 |
| 4 | **2a** (4.0) | L₃ (10) | DCM | 60 | 1.5 | 16 | 63 | 3 |
| 5 | **2a** (4.0) | L₃ (10) | DCE | 60 | 1.5 | 37 | 29 | 2 |
| 6 | **2a** (4.0) | L₃ (10) | DCP | 60 | 1.5 | 40 | 28 | 3 |
| 7 | **2a** (4.0) | L₃ (10) | DCB | 60 | 1.5 | 35 | 42 | 3 |
| 8 | **2a** (3.0) | L₃ (10) | DCE | 40 | 1 | 64 | 16 | 5 |
| 9 | **2a** (3.0) | L₁ (10) | DCE | 40 | 1 | 0 | 1 | 13 |
| 10 | **2a** (3.0) | L₂ (10) | DCE | 40 | 1 | 12 | 1 | 16 |
| 11 | **2a** (3.0) | L₄ (10) | DCE | 40 | 1 | 5 | 16 | 5 |
| 12 | **2a** (3.0) | L₃ (20) | DCE | 40 | 1 | 77 | 9 | 3 |
| 13 | **2b** (3.0) | L₃ (20) | DCE | 40 | 1 | 84 (78[b]) | 4 | 2 |
| 14 | **2a** (3.0) | L₅ (2.5) | DCE | 40 | 1 | 26[c] | Trace | 24 |
| 15 | **2a** (3.0) | L₆ (5) | DCE | 40 | 1 | 7[d] | Trace | 90 |
| 16 | **2a** (1.5) | L₆ (5) | DCE | 40 | 1 | 6 | Trace | 88 (80[b, d]) |

PhSiH₃   n-C₈H₁₇SiH₃   Ph₂P⌢PPh₂   Ph₂P⌢⌢PPh₂   Ph₂P⌢⌢⌢PPh₂   (PPh₂/PPh₂ benzene)   L₅ (xantphos)   L₆ (cy-xantphos)

**2a**   **2b**   L₁ (dppm)   L₂ (dppe)   L₃ (dppp)   L₄ (dppbz)   L₅ (xantphos)   L₆ (cy-xantphos)

*CuTc* copper(I) thiophene-2-carboxylate, *Cy-H* cyclohexane, *MTBE* methyl tert-butyl ether, *THF* tetrahydrofuran, *DCM* dichloromethane, *DCE* 1,2-dichloroethane, *DCP* 1,3-dichloropropane, *DCB* 1,4-dichlorobutane.
[a]Reaction conditions: CuTc, ligand, **1a** (0.2 mmol) and **2** were stirred in solvent (1.0 mL) under nitrogen atmosphere. The yields were determined by crude ¹H NMR using 1,1,2,2-tetrachlorethane as an internal standard.
[b]Isolated yield.
[c]CuTc = 2.5 mol%.
[d]CuTc = 5 mol%.

multiple-substituted enynes show low reactivity under this reaction condition, which may be due to the increased stereo hinderance of these substrates.

Chiral allenylsilanes are versatile building blocks in asymmetric synthesis[2,7,8]. Thus, we investigated the enantioselective preparation of allenylsilanes (Table 2). First, we evaluated different chiral phosphine ligands by using Cu(OAc)₂ as catalyst precursor in DCE. When L₁* which has a similar structure to dppp was applied as the chiral ligand, **6aa** was delivered in moderated yield but the enantioselectivity was nominal (entry 1). Other commonly used chiral phosphine ligands such as BINAP, SEGPHOS only dilivered the desired product in poor yields with low enantioselectivity induction (entries 2–4). When L₅* was used as the ligand, product **6aa** was obtained in 54% yield and 43% ee (entry 5). We were pleased to find that the ee value of the product was highly dependent on the reaction temperature (entries 5-10). When the reaction was run at −50 °C, the ee value of **6aa** could be enhanced to 96% (entry 10). Decreasing the loading of **2a** to 2.0 equivalents has no negative effect on the yield or ee value of **6aa** (entry 11).

Under the above optimized conditions, the substrate scope of enantioselective 1,4-hydrosilylation was investigated (Fig. 3). In general, all substrates produced the corresponding allenylsilane products in good yields with high enantioselectivities (up to 99% ee). The substrates bearing electronic-neutral groups exhibited good reactivity and

provided the target products in good yields with excellent enantioselectivities (**6ba-6ea**). Besides, a variety of functional groups such as halides, methoxy and acetal could be well tolerated (**6ga-6ka**). Thiophene- and indole-substituted 1,3-enynes also worked well. Furthermore, alkyl- or alkenyl- substituted substrates could be uneventfully transformed into the target products with excellent enantioselectivities (**6oa-6ra**). Besides, 3,5-dimethyl, -F, -Cl, and -CF₃ substituted phenyl silanes produced the corresponding products in good yields and with high ee values (**6ac-6af**). The absolute configuration of product **6ei** was determined by single-crystal X-ray diffraction analysis (CCDC no: 2183554).

The substrate scope of 1,2-hydrosilylation of 1,3-enynes was also investigated (Fig. 4). The substrates tethered with alkyl groups on the phenyl ring delivered the desired products in high yields and excellent regioselectivities (**5ba-5ea**). Naphthyl substituted substrates were also feasible (**5la, 5ma**). Various halogen atoms such as F, Cl, Br on different positions of the phenyl ring could be well tolerated (**5ga-5ka**). A collection of aryl substituted 1,3-enynes bearing with electron-donating or electron-withdraw groups at different positions of phenyl ring were converted to the corresponding propargylsilane products in satisfactory to high yields and with high regioselectivities (**5na-5ua**). Sterically encumbered 1-(but-3-en-1-yn-1-yl)naphthalene and *ortho*-substituted substrates reacted smoothly and provided the desired products in

higher regioselectivities (**5ba, 5ga, 5ia, 5la, 5ra**). Besides, highly reactive alkenyl and alkynyl moieties could also be well tolerated (**5wa-5za**). The reactivities of alkyl substituted 1,3-enynes were also examined. The corresponding propargylsilane product **5aba** was obtained in high regioselectivity, but with moderate efficiency even increasing the loading of catalyst and extending the reaction time, which was attributed to the lower reactivity of alkyl-substituted 1,3-enynes. With respect to the silanes, a series of trihydrosilanes are competent partners. A set of functional groups such as –F, –Cl, –OMe, –NMe₂ and –CF₃ could be well tolerated. Besides, *n*-octylsilane also showed high reactivity. However, dihydrosilanes such as PhMeSiH₂ and Ph₂SiH₂ gave poor results probably due to their larger steric hindrance.

We subsequently surveyed the asymmetric 1,2-hydrosilylation reaction of 1,3-enynes. When (*R*)-DTBM-SEGPHOS was used as chiral ligand, the corresponding chiral propargylsilane products were obtained in high yields with moderate enantioselectivities (Fig. 5). After extensive screening of various types of phosphine and nitrogen containing ligands and other factors, the enantioselectivity and regioselectivity still could not been well controlled simultaneously (see Supplementary Table 1 for details). Therefore, the possible chiral induction step was explored. When 0.75 equivalent H₂O was added, the disubstituted allene compound **9** was obtained in 15% yield with 58% ee value. This result indicated that when **2k** was used as the silane

source the moderate enantiomeric excess of chiral propargylsilane products may stem from the moderate enantioselectivity of the initial hydrocupration process of enyne with CuH species.

## Scale-up experiments and derivatizations
The practicality of this strategy was demonstrated by the scale-up experiments (Fig. 6a). Both racemic and asymmetric reaction delivered the corresponding products in high yields and without an obvious diminishment of enantioselectivity. The chiral allenylsilanes could be further converted to silanol or silyl ether products (**10, 11**) in moderate to high yields and without erosion of enantioselectivity[73,74]. Besides, the chiral allenylsilanes **6aa**, which serve as a chiral silanes source, reacted smoothly with phenyl allene and provided tertiary silane **12** in high yield with the retention of enantioselectivity[63]. Iodoarylation of *para*-methoxy substituted allenylsilanes delivered 2-iodo-3-silylindene **14** in good yield (Fig. 6b)[75].

## Mechanistic investigations
To shed light on the reaction mechanism, several deuterium-labeling reactions were performed. The reaction of (4-methoxyphenyl)silane-*d₃* with **1a** under standard conditions was surveyed, which provided deuterated products ***d*−3ag** and ***d*-5ah** in 65% and 87% yields (Fig. 7a-i/ii). Deuterium oxide quenching

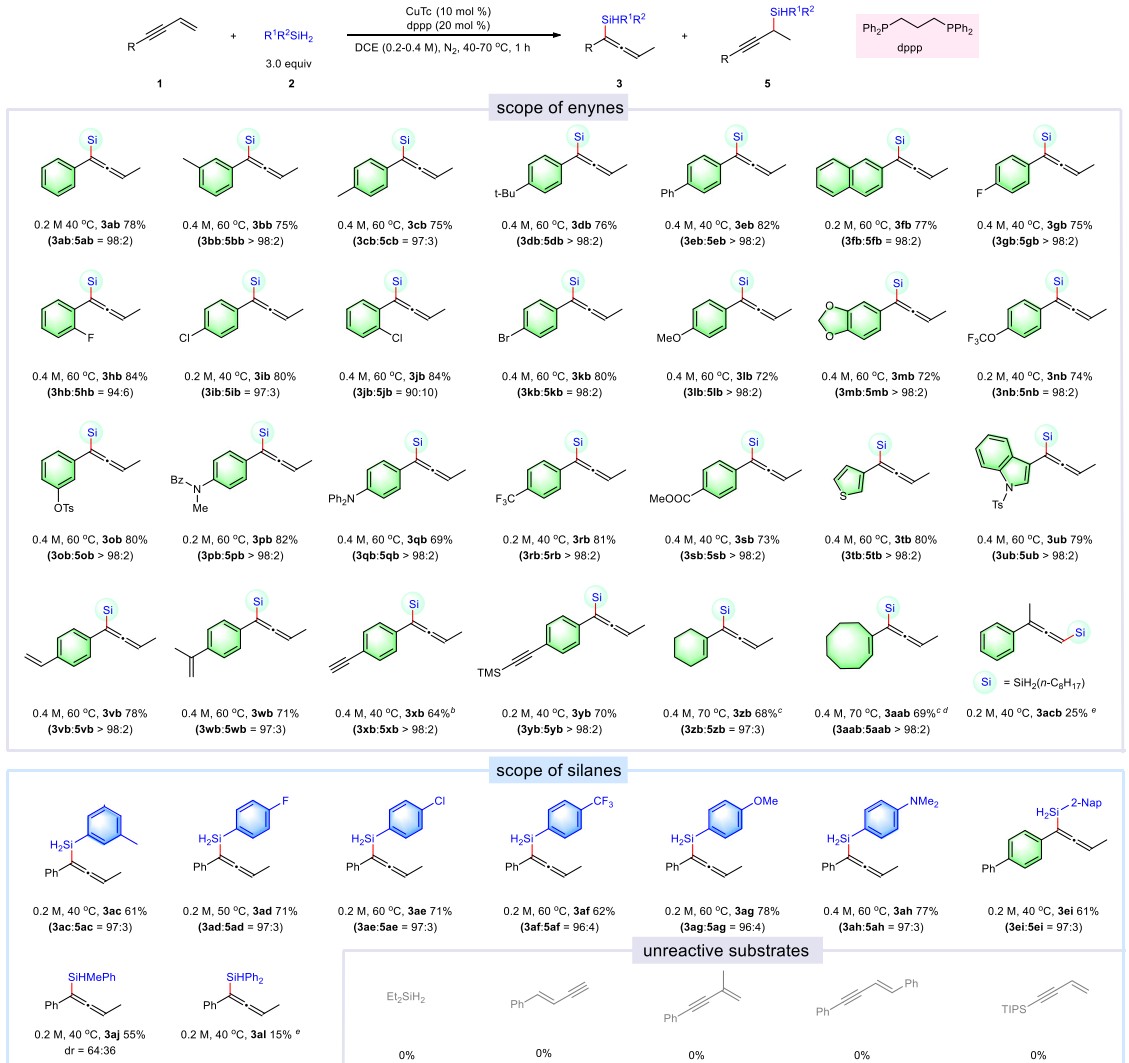

**Fig. 2 | Substrates scope of copper-catalyzed 1,4-hydrosilylation of 1,3-enynes with silanes[a].** [a]The mixture of **1** (0.2 mmol), R¹R²SiH₂ (0.6 mmol), CuTc (10 mol %), and dppp (20 mol %) in DCE (1.0 mL) was stirred for 1 h under nitrogen atmosphere in an oil bath. [b]t = 3 h, [c]dppp = 10 mol%, solvent = DCM, [d]t = 4 h, [e]¹H NMR yield.

## Table 2 | Copper-catalyzed asymmetric 1,4-hydrosilylation of 1,3-enynes[a]

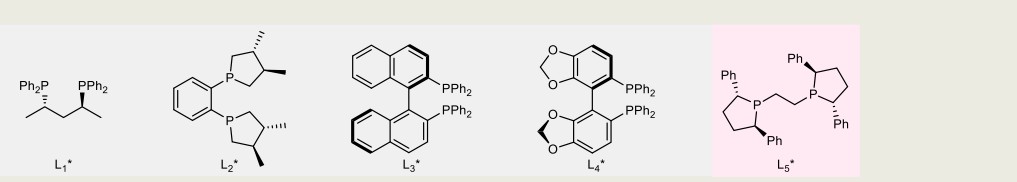

| entry | ligand (5 mol %) | T/ °C | t/h | 6aa | | 7aa |
|---|---|---|---|---|---|---|
| | | | | yield % | ee % | yield % |
| 1 | L₁* | 40 | 1 | 44 | 1 | trace |
| 2 | L₂* | 40 | 1 | 10 | −12 | 3 |
| 3 | L₃* | 40 | 1 | 24 | −19 | trace |
| 4 | L₄* | 40 | 1 | 10 | −32 | 1 |
| 5 | L₅* | 40 | 1 | 54 | 43 | trace |
| 6 | L₅* | 0 | 1 | 42 | 68 | trace |
| 7 | L₅* | −10 | 1 | 23 | 73 | trace |
| 8 | L₅* | −20 | 2 | 38 | 79 | trace |
| 9 | L₅* | −30 | 11 | 60 | 85 | trace |
| 10 | L₅* | −50 | 11 | 75[b] | 96 | trace |
| 11 | L₅* | −50 | 11 | 78 (72)[b,c,d] | 96 | trace |

[a]Reaction conditions: Cu(OAc)₂ (5 mol %), ligand (5 mol %), **1a** (0.2 mmol) and **2a** were stirred in DCE (1 mL) under nitrogen atmosphere, the yields were determined by crude ¹H NMR using 1,1,2,2-tetrachlorethane as an internal standard.
[b]**1a** and **2a** were added under −35 °C.
[c]DCE = 2 mL, PhSiH₃ = 2.0 equiv.
[d]Isolated yield.

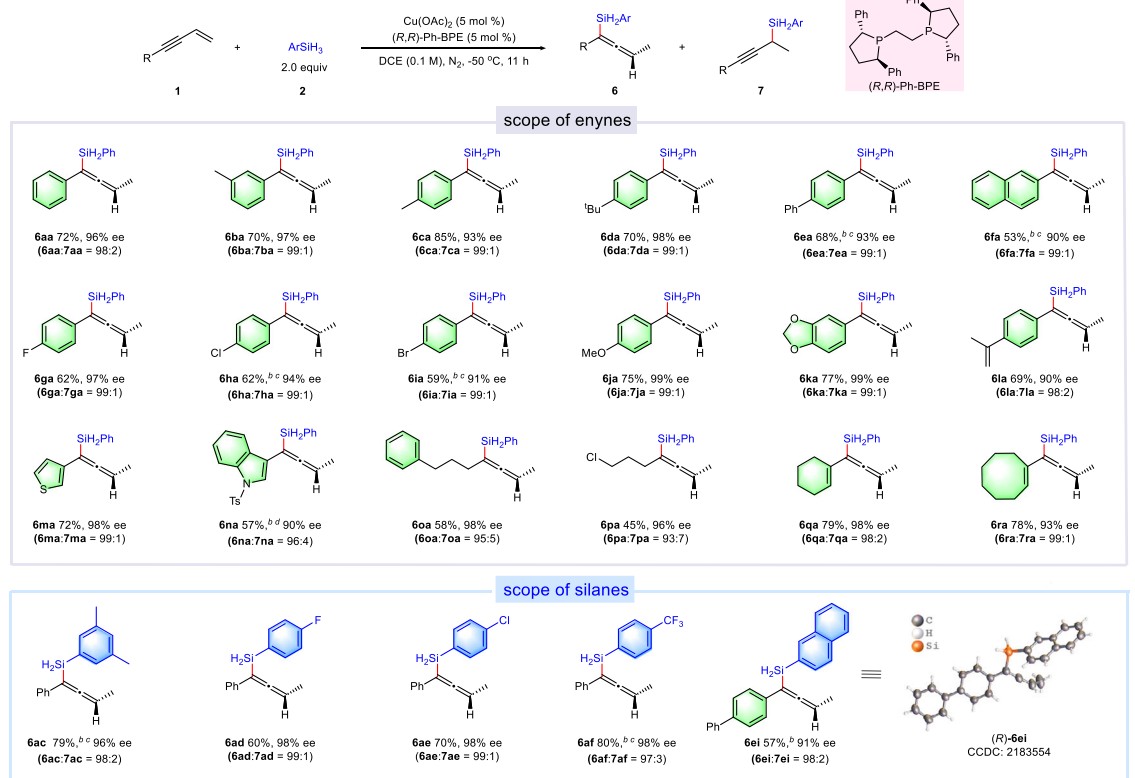

**Fig. 3 | Substrates scope of copper-catalyzed asymmetric 1,4-hydrosilylation of 1,3-enynes with silanes[a].** [a]The mixture of **1** (0.2 mmol), ArSiH₃ (0.4 mmol), Cu(OAc)₂ (5 mol %), and (R,R)-Ph-BPE (5 mol %) in DCE (2.0 mL) was stirred at −50 °C under nitrogen atomosphere. [b]**2** = 3.0 equiv, [c]t = 24 h, [d]t = 30 h.

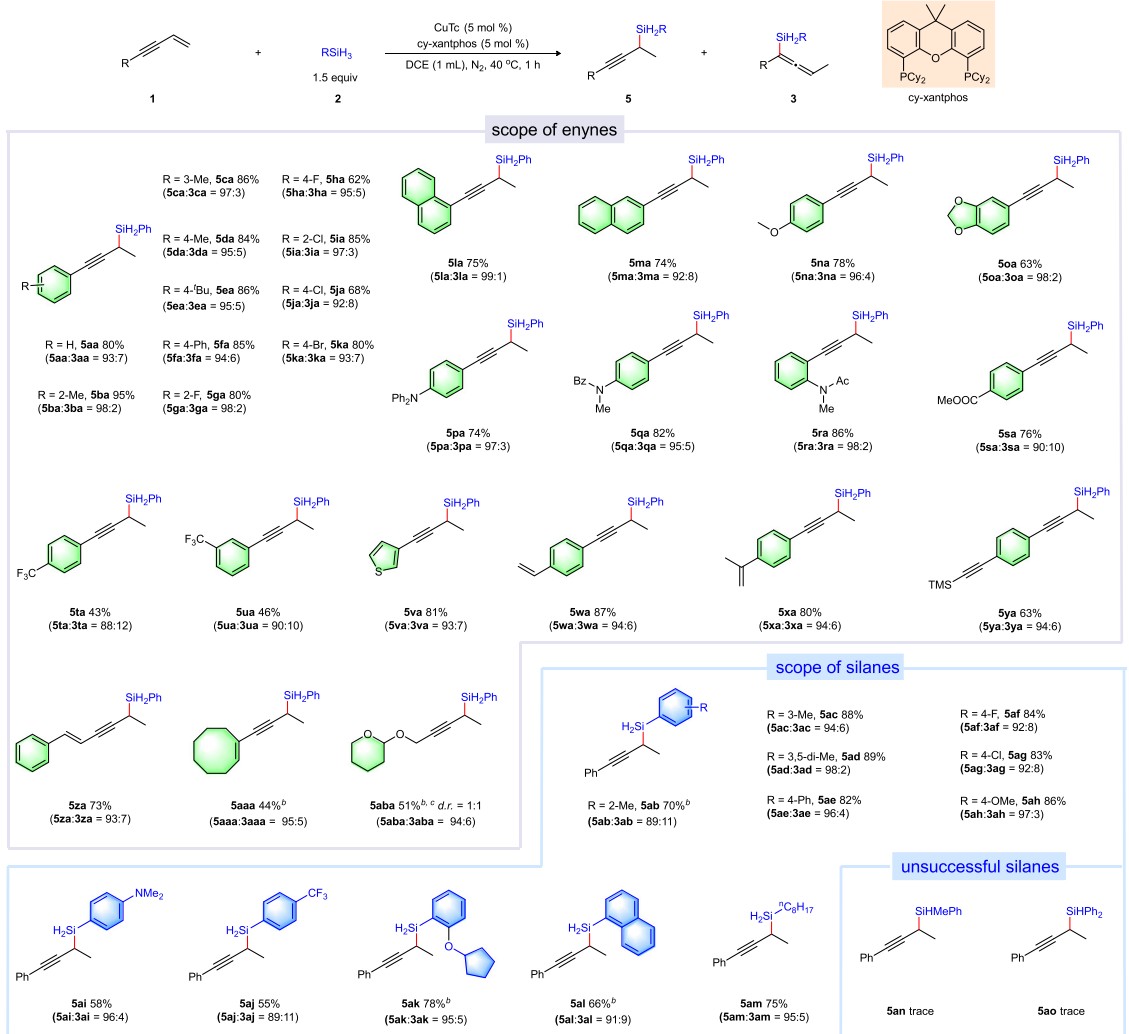

**Fig. 4 | Substrates scope of copper-catalyzed 1,2-hydrosilylation of 1,3-enynes with silanes[a].** [a]The mixture of **1** (0.2 mmol), RSiH$_3$ (0.3 mmol), CuTc (5 mol %) and cy-xantphos (5 mol %) in DCE (1.0 mL) was stirred at 40 °C for 1 h under nitrogen atmosphere. [b]CuTc = 10 mol %, cy-xantphos = 10 mol %, DCE = 0.5 mL. [c]t = 36 h.

experiments were also conducted. For the 1,4-hydrosilylation reaction, alongside the desired product **3ea**, deuterated allene **16** was obtained in 13% yield in the presence of 2.0 equivalent D$_2$O (Fig. 7a-iii). For the 1,2-hydrosilylation process, compounds **15** and **16** were also obtained when 0.75 equivalent D$_2$O was added (Fig. 7a-iv). The formation of **16** as the major side-product in both 1,2- and 1,4-hydrosilylation process suggests that allenylcopper might be the dominant reactive intermediate. Based on the results of these deuterium-labeling reactions, a simplified catalytic cycle was proposed (Fig. 7b). The incipient CuH species undergoes insertion with the enyne to form a propargylic Cu(I) species **I**, which may be in equilibrium with the allenylcopper **III**. From either **I** or **III**, metathesis with the silane could take place to form the final products and regenerate the CuH catalytic species.

To gain further mechanistic insights regarding the reactive intermediates and their reactivities, particularly the remarkable ligand effect on the regiocontrol, we performed density functional theory (DFT) calculations (Fig. 7c). Our calculations, congruent with the above experimental observation and calculations of Hoveyda[76] and Buchwald[77] and their respective coworkers, suggest that the corresponding allenylcopper **III** is more thermodynamically stable regardless of the supporting ligand, with a low or non-existing interconverting barrier (Fig. 7c), which is also consistent with the above experimental observation. Notably, although the metathesis of allenylcopper with PhSiH$_3$ via 4- and 6-membered transition states are both feasible, in the dppp ligated pathway, the four-membered transition state **TS$^4_{dppp}$** is lower in energy by 5.3 kcal/mol, likely due to the favorable interaction energy of −46.6 kcal/mol, as revealed by distortion interaction analysis[78], in which a π-π stacking interaction between one P-aryl of the ligand and the Ph group of PhSiH$_3$ (Fig. 7d, region A) and the attractive dispersion between another P-aryl of the ligand and the terminal C≡CMe moiety of the allenylcopper (Fig. 7d, region B) may play an important role, according to interaction region indicator analysis[79] (Fig. 7d, i). Interestingly, our calculation also revealed that in the xantphos ligated pathway, the four-membered transition state **TS$^4_{xant}$** (17.5 kcal/mol) is energetically very close to the six-membered transition state **TS$^6_{xant}$** (16.6 kcal/mol), which agrees well with the experimental observation (Table 1, entry 14). In the cy-xantphos ligated pathway, the six-membered transition state **TS$^6_{cy-xant}$** is now more favored by 3.3 kcal/mol. This is because, while the barrier of 6-membered transition states remained almost not changed upon changing the ligand from dppp to cy-xantphos (from 19.6 to 19.0 kcal/mol), that of the 6-membered transition state is elevated by 8.0 kcal/mol (from 14.3 to 22.3 kcal/mol). As revealed in the interaction region indicator analysis in Fig. 7d, iv, **TS$^6_{cy-xant}$** features more extensive dispersion interaction, especially between the phenyl ring of PhSiH$_3$ and the backbone of cy-xantphos, which counteracts the unfavorable endocyclic strain that was observed in both six-membered transition structures (Fig. 7d, regions C and G). This is also supported by the greater overall distortion energy and interaction in both components

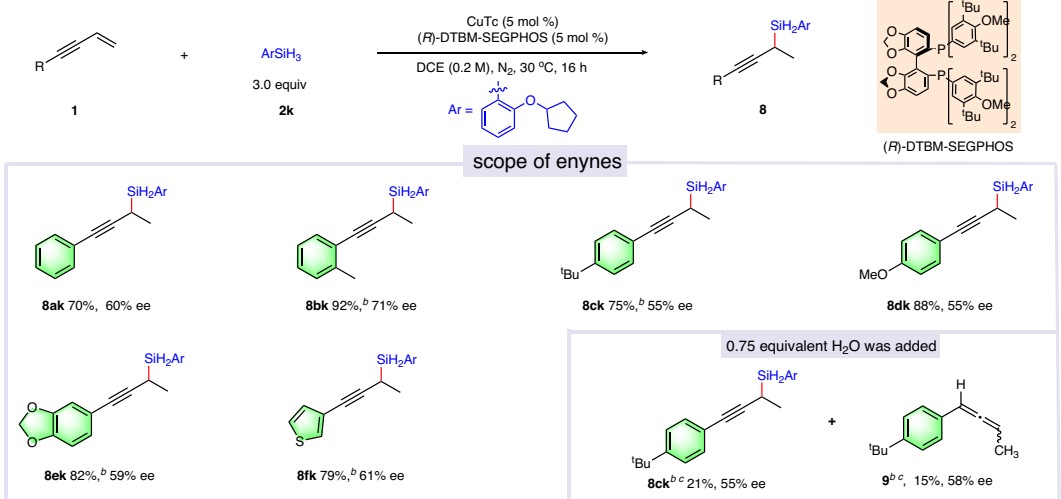

**Fig. 5 | Selected examples of copper-catalyzed asymmetric 1,2-hydrosilylation of 1,3-enynes with silanes[a].** [a]The mixture of **1** (0.2 mmol), **2k** (0.6 mmol), CuTc (5 mol %) and (*R*)-DTBM-SEGPHOS (5 mol %) in DCE (1.0 mL) was stirred at 30 °C under nitrogen atmosphere. [b]t = 24 h. [c]H NMR yield.

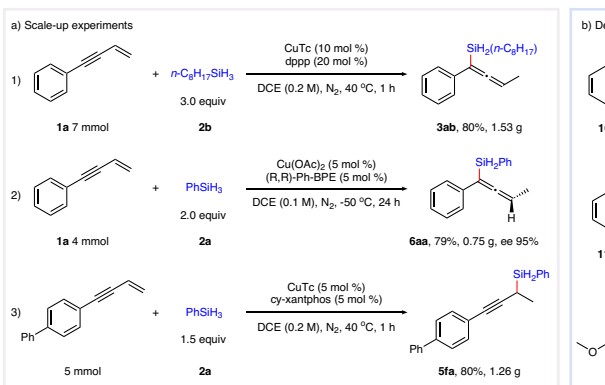

**Fig. 6 | Scale-up experiments and derivatizations. a** Scale-up experiments. **b** Derivatizations of products. (i) [RuCl₂(*p*-cymene)]₂ (1 mol %), MeOH (0.2 M), N₂, 10 min; (ii) H₂O₂ (2.0 equivalent), KHCO₃ (0.3 equivalent), THF/MeOH (1/1), N₂, 25 min. iii) CuTc (10 mol %), cy-xantphos (10 mol %), DCM, 30 °C, 4 h; iv) NIS (1.2 equivalent), CH₃CN (0.05 M), N₂, 30 °C, 20 min.

of **TS⁶_cy-xant** (as compared to those of **TS⁶_dppp**) which are canceled to roughly the same degree by the greater interaction energy, making the final barrier comparable to that of **TS⁶_dppp**. In contrast, the four-membered transition structure, **TS⁴_cy-xant**, has a high energy due to the presence of only sporadic dispersion interactions besides region E. Overall, it can be concluded that the observed regioselectivity in both cases are a result of more favorable dispersion effect of the ligand exerted to either of the two reacting partners. In addition, tentative location of the transition states from the higher energy propargyl copper intermediate **I** was also attempted, but they are found to be much higher in energy for the dppp ligated pathway (see Supplementary information for details, Supplementary Fig 3), in consistent with Hoveyda and coworkers' observation[76], so they are not further considered.

The enantioselective 1,4-hydrosilylation process was also studied by theoretical calculation with (*R, R*)-Ph-BPE as the ligand. Due to the weak interaction between the allenyl copper and silane moieties, and the resulted high flexibility of the latter in the metathesis event, the trajectories of silane were systematically mapped, and four trajectories were located for both (*R*)- and (*S*)-antipodes (see Supplementary Tab 4). Of these conformers **TS_M_conf1** and **TS_M_conf2** are very similar in geometry and differ only at the conformations of phenyl moiety of the silane. To obtain accurate energies for quantitative analysis and obviate the errors caused by the choice of functionals, electronic energy calculation was performed with Orca 5.0[80,81] using domain-based local pair natural orbital coupled-cluster with single, double, and perturbative triples excitation (DLPNO-CCSD(T)) method[82]. The energies of these TSs (Fig. 7e, i) shows a clear favorability to the observed (*R*)-configuration, with **TS_M_conf1_R** being the most dominant conformer, accounting to 62% product, although **TS_M_conf2_R** and **TS_M_conf3_R** also made nonnegligible contribution (20% and 13%, respectively). The most contributing conformers for (*S*)-antipode are **TS_M_conf1_S** and **TS_M_conf2_S**, each accounting for only 1% of the total products. Boltzmann analysis of the energies shows an overall enantioselectivity of 94% ee, agreeing very well with experimental data (Table 2, entry 11, 96% ee). The lowest energy conformers, **TS_M_conf1** and **TS_M_conf2**, feature a silane phenyl moiety staying closer to the chiral ligand's two phenyl moieties and away from the Ph substituent of the enyne substrate (Fig. 7e, ii), in line with the experimental observation that the enantioselectivity is insensitive to substitutions on the enyne (Fig. 3). Comparison of the geometric parameters of **TS_M_conf1_R** and **TS_M_conf1_S** (Fig. 7e, ii) revealed that the former is an earlier transition state featuring a slightly shorter C–Cu (2.10 vs 2.12 Å) and Si–H bond (1.56 vs 1.57 Å), as well as a significantly longer C–Si bond (2.33 vs 2.25 Å). Meanwhile, the distances between Cu and H and Cu and Si atoms are only very slightly shorter in **TS_M_conf1_R**,

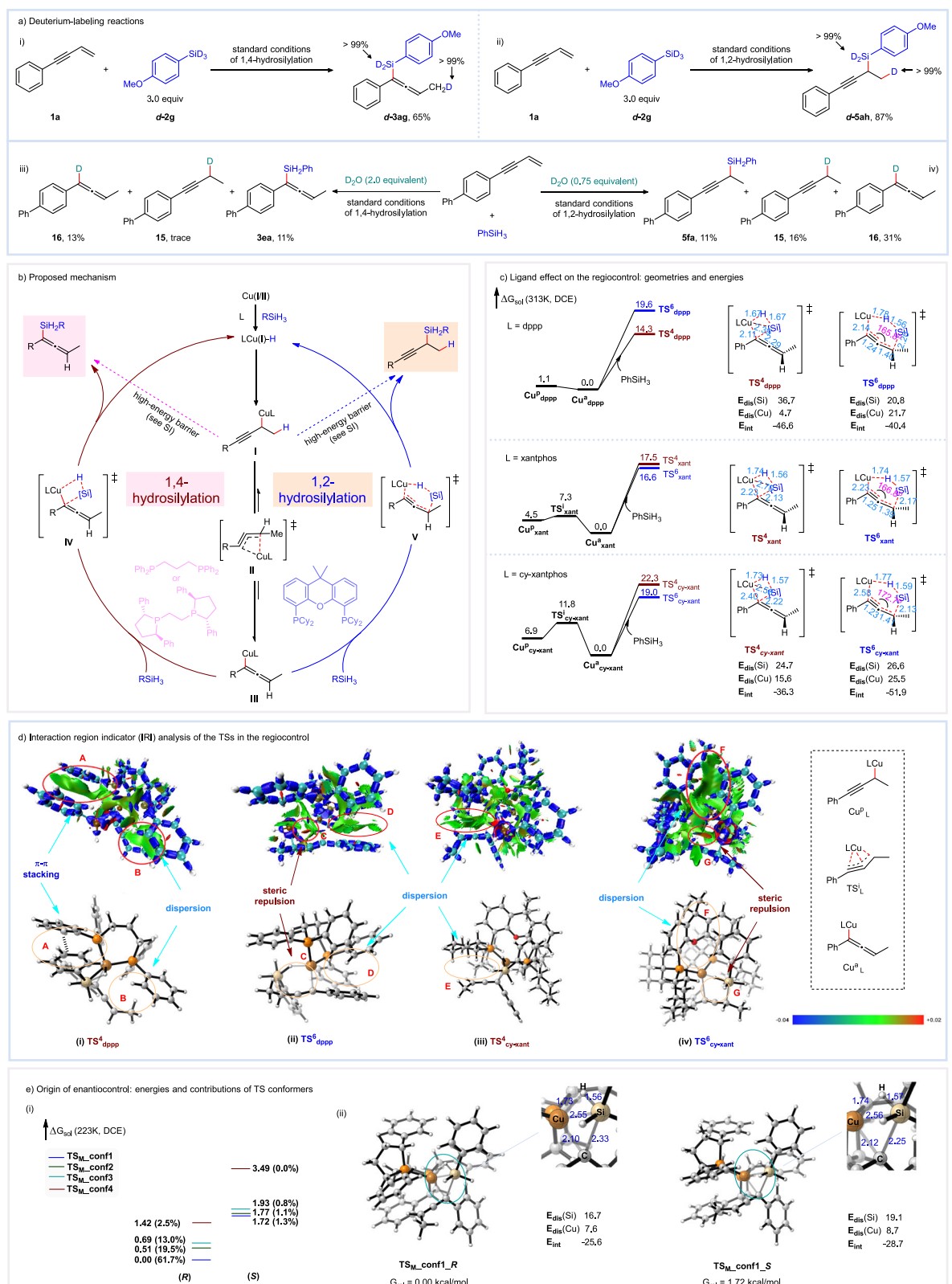

**Fig. 7 | Theoretical calculation guided mechanistic proposal and origin of selectivity control. a** Deuterium-labeling reactions. **b** Proposed mechanism. **c** Free energy profile of propargylic/allenylic copper and their metathesis with PhSiH$_3$ with different supporting ligands, the key geometrical parameters of the TSs and the distortion-interaction analysis. Level of theory: MN15-def2-tzvpp:SMD(DCE)//PBE0-D3(BJ)/6−31 G(d)/SDD(Cu):SMD(DCE). **d** Interaction region indicator (IRI) analysis of the competing transitions structures with dppp and cy-xantphos as ligands.

Isosurface value: 0.5. **e-i** Energies of the metathesis TS conformers in the enantio-selective 1,4-hydrosilylation and their Boltzmann distribution dictated contribution. **e-ii** Structures of the dominating pair of metathesis TSs and their distortion/ interaction analysis. Level of theory: DLPNO-CCSD(T)/cc-pVTZ:SMD(DCE)//PBE0-D3(BJ)/6−31G(d)/SDD(Cu):SMD(DCE). Bond length are in Å and energies in kcal/mol.

precluding a significantly better interaction. Distortion interaction analysis corroborates the above results, validating that the favoring of the *R*-configuration in the product is caused by less distortion in both the allenyl copper and silane fragments, rather than a more favorable interaction term.

## Discussion

We have realized the regio-controllable 1,2- and 1,4-hydrosilylation of 1,3-enynes by copper-based catalytic system wherein the regioselectivity is regulated by the choice of ligands. This reaction exhibits broad substrate scope. Both alkyl-, vinyl- and (hetero)aryl-substituted 1,3-enynes provided the corresponding propargylsilane and allenylsilane products in moderate to high yields. To our knowledge, this tactic represents the first example of selective 1,2-hydrosilylation of 1,3-enynes. The asymmetric 1,2-hydrosilylation was also studied preliminarily. In addition, as the first example of earth-abundant metal catalyzed asymmetric 1,4-hydrosilylation of 1,3-enynes excellent enantioselectivities (up to 99% ee) were realized under mild conditions. Density functional theory (DFT) calculation pinpointed the structural features of the ligand on the metathesis transition states responsible for the regiodivergence. High level DLPNO-CCSD(T) calculation on the enantioselective 1,4-hydrosilylation process also revealed the mechanism for enantio-differentiation in the four-membered metathesis step and the flexible nature of the silane during metathesis with organocopper (I) species. These studies can help further improve relevant processes.

## Methods

General procedure for racemic 1,4-hydrosilylation reaction: An oven dried 10-mL Schlenk tub equipped with a stirring bar was charged with CuTc (10 mol %, 0.02 mmol, 3.8 mg), dppp (20 mol %, 0.04 mmol, 16.5 mg), extra dry DCE (0.5 or 1.0 mL), **2** (3.0 equiv, 0.6 mmol) and **1** (0.2 mmol) in sequence. The reaction mixture was stirred at the indicated temperature for 1 h. Then, ethyl acetate was added, and the precipitate was removed by filtration. The resultant solution was concentrated, and the crude product was purified by column chromatography.

General procedure for enantioselective 1,4-hydrosilylation reaction: An oven dried 10-mL Schlenk tube equipped with a stirring bar was added Cu(OAc)$_2$ (5 mol %, 0.01 mmol, 1.8 mg), (*R*,*R*)-Ph-BPE (5 mol %, 0.01 mmol, 5.1 mg) and 2 mL extra dry DCE. The mixture was stirred at 30 °C for 5 minutes, then cooled to −35 °C (about 1 minute) following by adding **2** (2.0 equiv, 0.4 mmol) and **1** (0.2 mmol) to it at the same temperature. After that, the reaction mixture was stirred at −50 °C. Then ethyl acetate was added, and the precipitate was removed by filtration. The resultant solution was concentrated, and the crude product was purified by column chromatography.

General procedure for racemic 1,2-hydrosilylation reaction: An oven dried 10-mL Schlenk tube equipped with a stirring bar was added CuTc (5 mol %, 0.01 mmol, 1.9 mg), cy-xantphos (5 mol %, 0.01 mmol, 6.0 mg), 1 mL extra dry DCE, **2** (1.5 equiv, 0.3 mmol) and **1** (0.2 mmol) in sequence. The reaction mixture was stirred at 40 °C. After completion, ethyl acetate was added, and the precipitate was removed by filtration. The resultant solution was concentrated, and the crude product was purified by column chromatography.

General procedure for enantioselective 1,2-hydrosilylation reaction: An oven dried 10-mL Schlenk tube equipped with a stirring bar was added CuTc (5 mol %, 0.01 mmol, 1.9 mg), (*R*)-DTBM-SEGPHOS (5 mol %, 0.01 mmol, 11.8 mg) and 1 mL extra dry DCE. The mixture was stirred at 30 °C for 30 minutes, then **2k** (3.0 equiv, 0.6 mmol, 115.4 mg) and **1** (0.2 mmol) were added. After that, the reaction mixture was stirred at 30 °C. After completion, ethyl acetate was added, and the precipitate was removed by filtration. The resultant solution was

concentrated, and the crude product was purified by column chromatography.

## Data availability

All data generated or analyzed during this study are included in this published article (and its Supplementary information file). For the experimental procedures, data of NMR and HPLC analysis, see Supplementary information file. The crystallographic data of compound **6ei** is available at Cambridge Crystallographic Data Centre under the deposition number CCDC: 2183554. These data can be obtained free of charge from The Cambridge Crystallographic Data Center via www.ccdc.cam.ac.uk/ data_request/cif.

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

## Acknowledgements

The authors gratefully acknowledge research support of this work by the funding of the National Natural Science Foundation of China (21871240, (Y.-H.X.), 21871045, (J.-B.Z.)), the State Key Laboratory of Elemento-organic Chemistry Nankai University (202001, (Y.-H.X.)), the Fundamental Research Funds for the Central Universities (WK2060000017, (Y.-H.X.), and the Open Project of Key Laboratory of Organosilicon Chemistry and Material Technology of Ministry of Education, Hangzhou Normal University (KFJJ2022013, (Y.-H.X.)).

## Author contributions

Y.-H.X. and Z.-L.W. conceived the project. Z.-L.W., Q.L., M.-W.Y., Z.-X.S., Z.-Y.X., and W.-W.M. performed the experiments and analyzed the data. J.-B.Z. conducted the DFT calculations. Y.-H.X. supervised the research. Z.-L.W. and Y.-H.X. co-wrote the manuscript.

## Competing interests

The authors declare no competing interests.
