## [Peer Review File · Nature Communications]

Regio- and Enantioselective CuH-Catalyzed 1,2- and 1,4-Hydrosilylation of 1,3-EnynesReviewers' Comments:

Reviewer #1:

Remarks to the Author:

In this manuscript, the authors describe a ligand controlled Cu-catalyzed regiodivergent 1,2- and 1,4-hydrosilylation of 1,3-enynes. This method offers a facile access to various allenyl- and propargyl silanes with high yields and excellent regioselectivity. Impressively, the rational choice of chiral ligand allows the asymmetric 1,4-hydrosilylation to afford axially chiral allenylsilanes in high enantioselectivity. The origin of the regioselectivity and the effect of ligands were studied by DFT calculation. The full manuscript and the supporting information were well prepared. Publication is recommended after minor revisions:

1. The substrate scope is limited to 4-substituted 1,3-enynes. The impact of substitution pattern of 1,3-enynes on the hydrosilylation should be investigated. For example, 4-silyl-1,3-enynes, 1- or 2-substituted 1,3-enynes. It would be helpful for the readers if a few structures of representative unsuccessful examples were shown.
2. The title of Table 3 is the same with Table 1, which is not correct. The content of table 3 is not 'regiodivergent' but 'asymmetric 1,4-hydrosilylation'.
3. In table 3, the yields of 6aa decreased as the temperature was changed from 0 to -10 oC, but increased as the temperature ranged from -10 to -50 oC (entries 6-11). The authors are suggested to provide explanation for this phenomena. How about the conversion or selectivity for these reactions?

Reviewer #2:

Remarks to the Author:

The article by Xu, Zhao and co-workers describes a regioselective, Cu-catalyzed hydrosilylation of enynes in which the regioselectivity appears to be controlled by the ligand. The levels of regioselectivity in these processes (1,2- and 1,4-hydrosilylation, the latter reported in the scientific literature) are very good in most cases, and the yields also acceptable for both transformation. The reaction is catalyzed by Cu, which is a step towards more sustainable reactions.

However, I find the majority of the results reported in this draft lack the novelty required for a journal such as Nature Communications for the following reasons:

1. A significant limitation in both reactions, as they stand, pertains to substitution at the enyne, which appears to be restricted to modifications at the alkyne terminus. Why is this the case?
2. Perhaps more importantly, the levels of asymmetric induction are only good in the 1,4-hydrosilylation process, whereas only a few examples of the 1,2-hydrosilylation reaction are reported in the manuscript.
3. In terms of derivatizations, only two reactions are described, and both of them deal with modifications of the silane. The authors should make an effort in this regard to showcase the utility of the reaction products, for example in Denmark couplings...
4. Finally, I do not find the DFT studies to be particularly striking. Although the computed attractive dispersion interactions appear to be important for the regio selective process, no details are given on the enantioselective transformation. Also, as the authors point out, the intermediacy of an allenyl-Cu species has already been demonstrated by Buchwald and Hoveyda.

Altogether, I do not think this report merits publication in this journal. I would recommend the authors

a more specialized venue, such as Organic Letters or the Journal of Organic Chemistry.

Reviewer #3:

Remarks to the Author:

This present manuscript by Xu and coworkers describes a Cu-catalyzed regio- and enantioselective 1,2- and 1,4-hydrosilylation of 1,3-enynes. Cu-H is regarded as the reactive species to initiate the reaction. The authors demonstrate their method enables both 1,2- and 1,4- hydrosilylation with excellent levels of regioselectivity by ligand-controlled strategy. More importantly, copper catalyzed enantioselective 1,4- hydrosilylation of 1,3-enynes was realized for the first time, giving the enantioenriched allenylsilanes in excellent ee values. Unfortunately, asymmetric 1,2- hydrosilylation was not successfully, only moderate ee could be observed.

In addition to scope, a series of experiments are conducted in order to gain mechanistic insight, including DFT calculation studies. For these studies, the authors proposed allenylcopper is probably the reactive intermediate. Along with manuscript the authors provide a detailed supporting information that is high quality and well done.

Overall, the authors have developed a good catalytic method for regioselective preparation allenylsilanes and propargylsilanes, which will be broad of interest to those in the fields of synthesis and catalysis as well as readers of Nature Communications. Consequently, I recommend this communication be accepted for publication with minor revisions.

- 1) In figure 1a, 1,4-hydrosilylation is missing. In Cu's work shown in figure 1b, Re should be replaced by La. In figure 1c, ee value was not given for asymmetric 1,2-hydrosilylation.
- 2) In table 1 and table 3, symbol % was given in columns of yield and ee, which make the table not concise. Symbol % should be only given in the first row.
- 3) From line 111 to 119, the description of the performance of L2 to L4 should added as well as the results of entry 11 in table 3. Otherwise, entry 11 could be removed from the table.
- 4) In line 162, the authors mentioned nitrogen containing ligands were screened. But, there was no N-containing ligands shown in Table S1. The very special silane was used in asymmetric 1,2-hydrosilylation, please explain. If possible, please include more detailed optimization data in SI.
- 5) To demonstrate the potential application of the products, the authors should give more examples of derivatization.

Reviewer #4:

Remarks to the Author:

Zhao, Xu and colleagues describe the development of regio- and enantioselective 1,2- and 1,4-hydrosilylation of 1,3-enynes catalyzed by copper-diphosphine complexes. The regioselectivity is ligand-dependent and uniquely prefers a 1,2-hydrosilylation when Cu-DTBM-SEGPHOS or Cu-Cy-Xantphos is applied. Overall, this work represents a notable addition to the current 1,4-hydrosilylation of 1,3-enynes as well as silane synthesis in general. Publication of this manuscript in Nature Communications may be supported, while the authors are advised to address the following points.

1. The ees are only moderate for the 1,2-hydrosilylations, mostly around 60%, under the circumstance of a sophisticated silane reagent. The enantioselectivity need to be improved with normal silanes.
2. The authors conclude that the dispersion effects between ligands and reacting reagents determine the regioselectivity. How other factors, such as pi-pi interactions and steric hindrance, are excluded from this circumstance?
3. Only reactions with RSiH₃ sources are provided. How are the reactions for silanes with two R groups?
4. The result for 1,4-hydrosilylation of ArSiD₃ and but-3-en-1-yn-1-ylbenzene (1a) is also required. The deuterium-labeling reaction part should be moved to the mechanistic study section.
5. For Table 1 & 3, CuTc and Cu(OAc)₂ are metal precursors, but not catalysts. Also, since only CuTc

and Cu(OAc)₂ are used in Table 1 & 3, the corresponding columns should be removed to simplify the Tables. The DCE column for Table 3 should also be removed.

6. Please provide the clear chemical structures along with the pictures for Fig 3c.

7. Other errors: For Fig 1a, three 1,2-hydrosilylations are present. The reference numbering for "... Hoveyda and coworkers' observation,⁷⁷" is not correct.

Comments

Reviewer #1 (Remarks to the Author):

In this manuscript, the authors describe a ligand controlled Cu-catalyzed regiodivergent 1,2- and 1,4-hydrosilylation of 1,3-enynes. This method offers a facile access to various allenyl- and propargyl silanes with high yields and excellent regioselectivity. Impressively, the rational choice of chiral ligand allows the asymmetric 1,4-hydrosilylation to afford axially chiral allenylsilanes in high enantioselectivity. The origin of the regioselectivity and the effect of ligands were studied by DFT calculation. The full manuscript and the supporting information were well prepared. Publication is recommended after minor revisions:

1. The substrate scope is limited to 4-substituted 1,3-enynes. The impact of substitution pattern of
2. The title of Table 3 is the same with Table 1, which is not correct. The content of table 3 is not 'regiodivergent' but 'asymmetric 1,4-hydrosilylation'.
3. In table 3, the yields of **6aa** decreased as the temperature was changed from 0 to -10 °C, but increased as the temperature ranged from -10 to -50 °C (entries 6-11). The authors are suggested to provide explanation for this phenomena. How about the conversion or selectivity for these reactions?

Reviewer #2 (Remarks to the Author):

The article by Xu, Zhao and co-workers describes a regioselective, Cu-catalyzed hydrosilylation of enynes in which the regioselectivity appears to be controlled by the ligand. The levels of regioselectivity in these processes (1,2- and 1,4-hydrosilylation, the latter reported in the scientific literature) are very good in most cases, and the yields also acceptable for both transformation. The reaction is catalyzed by Cu, which is a step towards more sustainable reactions. However, I find the majority of the results reported in this draft lack the novelty required for a journal such as Nature Communications for the following reasons:

1. A significant limitation in both reactions, as they stand, pertains to substitution at the enyne, which appears to be restricted to modifications at the alkyne terminus. Why is this the case?
2. Perhaps more importantly, the levels of asymmetric induction are only good in the 1,4-hydrosilylation process, whereas only a few examples of the 1,2-hydrosilylation reaction are reported in the manuscript
3. In the terms of derivatizations, only two reactions are described, and both of them deal with modifications of the silane. The authors should make an effort in this regard to showcase the utility of the reaction products, for example in Denmark couplings...

4. Finally, I do not find the DFT studies to be particularly striking. although the computed attractive dispersion interactions appear to be important for the regio selective process, no details are given on the enantioselective transformation. Also, as the authors point out, the intermediacy of an allenyl-Cu species has already been demonstrated by Buchwald and Hoveyda.

Altogether, I do not think this report merits publication in this journal. I would recommend the authors a more specialized venue, such as *Organic Letters* or the *Journal of Organic Chemistry*.

Reviewer #3 (Remarks to the Author):

This present manuscript by Xu and coworkers describes a Cu-catalyzed regio- and enantioselective 1,2- and 1,4-hydrosilylation of 1,3-enynes. Cu-H is regarded as the reactive species to initiate the reaction. The authors demonstrate their method enables both 1,2- and 1,4-hydrosilylation with excellent levels of regioselectivity by ligand-controlled strategy. More importantly, copper catalyzed enantioselective 1,4- hydrosilylation of 1,3-enynes was realized for the first time, giving the enantioenriched allenylsilanes in excellent ee values. Unfortunately, asymmetric 1,2- hydrosilylation was not successful, only moderate ee could be observed. In addition to scope, a series of experiments are conducted in order to gain mechanistic insight, including DFT calculation studies. For these studies, the authors proposed allenylcopper is probably the reactive intermediate. Along with manuscript the authors provide a detailed supporting information that is high quality and well done.

Overall, the authors have developed a good catalytic method for regioselective preparation allenylsilanes and propargylsilanes, which will be broad of interest to those in the fields of synthesis and catalysis as well as readers of Nature Communications. Consequently, I recommend this communication be accepted for publication with minor revisions.

- 1) In figure 1a, 1,4-hydrosilylation is missing. In Cui's work shown in figure 1b, Re should be replaced by La. In figure 1c, ee value was not given for asymmetric 1,2-hydrosilylation.
- 2) In table 1 and table 3, symbol % was given in columns of yield and ee, which make the table not concise. Symbol % should be only given in the first row.
- 3) From line 111 to 119, the description of the performance of L₂ to L₄ should added as well as the results of entry 11 in table 3. Otherwise, entry 11 could be removed from the table.
- 4) In line 162, the authors mentioned nitrogen containing ligands were screened. But, there was no N-containing ligands shown in Table S1. The very special silane was used in asymmetric 1,2-hydrosilylation, please explain. If possible, please include more detailed optimization data in SI.
- 5) To demonstrate the potential application of the products, the authors should give more examples of derivatization.

Reviewer #4 (Remarks to the Author):

Zhao, Xu and colleagues describe the development of regio- and enantioselective 1,2- and 1,4-hydrosilylation of 1,3-enynes catalyzed by copper-diphosphine complexes. The regioselectivity is ligand-dependent and uniquely prefers a 1,2-hydrosilylation when Cu-DTBM-SEGPHOS or Cu-cy-xantphos is applied. Overall, this work represents a notable addition to the current 1,4-hydrosilylation of 1,3-enynes as well as silane synthesis in general. Publication of this manuscript in Nature Communications may be supported, while the authors are advised to address the following points.

1. The ees are only moderate for the 1,2-hydrosilylations, mostly around 60%, under the circumstance of a sophisticated silane reagent. The enantioselectivity need to be improved with normal silanes.
2. The authors conclude that the dispersion effects between ligands and reacting reagents determine the regioselectivity. How other factors, such as π - π interactions and steric hindrance, are excluded from this circumstance?
3. Only reactions with RSiH₃ sources are provided. How are the reactions for silanes with two R groups?
4. The result for 1,2-hydrosilylation of ArSiD₃ and but-3-en-1-yn-1-ylbenzene (**1a**) is also required. The deuterium-labeling reaction part should be moved to the mechanistic study section.
5. For Table 1 & 3, CuTc and Cu(OAc)₂ are metal precursors, but not catalysts. Also, since only CuTc and Cu(OAc)₂ are used in Table 1 & 3, the corresponding columns should be removed to simplify the Tables. The DCE column for Table 3 should also be removed.
6. Please provide the clear chemical structures along with the pictures for Fig 3c.
7. Other errors: For Fig 1a, three 1,2-hydrosilylations are present. The reference numbering for "... Hoveyda and coworkers' observation,⁷⁷" is not correct.

Point-by-point response

Reviewer #1 (Remarks to the Author):

In this manuscript, the authors describe a ligand controlled Cu-catalyzed regiodivergent 1,2- and 1,4-hydrosilylation of 1,3-enynes. This method offers a facile access to various allenyl- and propargyl silanes with high yields and excellent regioselectivity. Impressively, the rational choice of chiral ligand allows the asymmetric 1,4-hydrosilylation to afford axially chiral allenylsilanes in high enantioselectivity. The origin of the regioselectivity and the effect of ligands were studied by DFT calculation. The full manuscript and the supporting information were well prepared. Publication is recommended after minor revisions:

1. The substrate scope is limited to 4-substituted 1,3-enynes. The impact of substitution pattern of 1,3-enynes on the hydrosilylation should be investigated. For example, 4-silyl-1,3-enynes, 1- or 2-substituted 1,3-enynes. It would be helpful for the readers if a few structures of representative unsuccessful examples were shown.

Response: Thanks for your suggestion. We have examined different substitution pattern of 1,3-enynes and these samples have been provided in the revised manuscript. Unfortunately, 1-substituted 1,3-enynes and 1,4-/2,4-disubstituted enynes are inert under the standard conditions and 2-substituted 1,3-enynes only delivered the desired product in 25% yield. These results may be due to the increased stereo hinderance of these substrates. In one side, the increased stereo hinderance of these substrates hamper the effective coordination of enynes with copper center. On the other side, the formation of propargylic Cu(I) intermediate initiates this reaction, but the increased stereo hinderance of these substrates disfavors the formation of the congested propargylic Cu(I) intermediate.

2. The title of Table 3 is the same with Table 1, which is not correct. The content of table 3 is not 'regiodivergent' but 'asymmetric 1,4-hydrosilylation'.

Response: Thanks for your reminding. We are sorry for these mistakes; we have corrected it now.

3. In table 3, the yields of **6aa** decreased as the temperature was changed from 0 to -10 °C, but increased as the temperature ranged from -10 to -50 °C (entries 6-11). The authors are suggested to provide explanation for this phenomenon. How about the conversion or selectivity for these reactions?

Response: The reaction efficiency decreased when lowering the reaction temperature. So, longer reaction time was used to obtain good conversion of **1a** when the reaction was performed at low temperature. When the reaction was performed at 0 °C or -10 °C, the reaction time was only 1 h, there is still some 1,3-enyne left in the reaction mixture. Therefore, longer reaction time (11 h) was used when the reaction temperature ranged from -10 to -50 °C. During the screening of asymmetric 1,4-hydrosilylation reaction, only trace amount of product **7aa** was observed and these results were provided in the revised manuscript, but some other side reactions are still present in some cases (such as reduction reaction, further hydrosilylation of **6aa** and others).

Reviewer #2 (Remarks to the Author):

The article by Xu, Zhao and co-workers describes a regioselective, Cu-catalyzed hydrosilylation of enynes in which the regioselectivity appears to be controlled by the ligand. The levels of regioselectivity in these processes (1,2- and 1,4-hydrosilylation, the latter reported in the scientific literature) are very good in most cases, and the yields also acceptable for both transformations. The reaction is catalyzed by Cu, which is a step towards more sustainable reactions. However, I find the majority of the results reported in this draft lack the novelty required for a journal such as Nature Communications for the following reasons:

1. A significant limitation in both reactions, as they stand, pertains to substitution at the enyne, which appears to be restricted to modifications at the alkyne terminus. Why is this the case?

Response: Thanks for your questions. We have examined different substitution patterns of 1,3-enynes, but unfortunately, other types of enynes such as 1- or 2-substituted 1,3-enynes or multiple-substituted enynes show low reactivity under these reaction conditions. These results may be ascribed to the increased steric hindrance of these substrates. On one side, the increased steric hindrance of these substrates hampers the effective coordination of the enyne with the copper center. On the other side, the formation of propargylic Cu(I) intermediate initiates this reaction, but the increased steric hindrance of these substrates disfavors the formation of the congested propargylic Cu(I) intermediate.

2. Perhaps more importantly, the levels of asymmetric induction are only good in the 1,4-hydrosilylation process, whereas only a few examples of the 1,2-hydrosilylation reaction are reported in the manuscript

Response: Thanks for your questions. Tremendous efforts have been devoted to improve the enantioselectivity of 1,2-hydrosilylation reaction (see revised supporting information for details), but only moderate asymmetric induction was obtained in this reaction. Therefore, only several representative examples were provided in the manuscript.

3. In the terms of derivatizations, only two reactions are described, and both of them deal with modifications of the silane. The authors should make an effort in this regard to showcase the utility of the reaction products, for example in Denmark couplings...

Response: Thanks for your questions. More efforts have been devoted to the derivatization of these products, and several results were listed on the revised manuscript. Among them, *para*-methoxy substituted allenylsilane can undergo iodoarylation reaction to construct 2-iodo-3-silylindene in the presence of NIS. The asymmetric iodoarylation reaction was also tried, but only racemic product was obtained. In addition, allenylsilane product can be used as chiral silane source for the allene hydrosilylation reaction. We searched different reaction patterns of allenylsilanes in SciFinder and Reaxys, but no Denmark coupling reaction of allenylsilanes was found. In spite of this, we tried different Denmark coupling reaction conditions of vinylsilanes and alkylsilanes, but no desired product was obtained.

unsuccessful Denmark coupling reactions and other derivatizations

4. Finally, I do not find the DFT studies to be particularly striking. Although the computed attractive dispersion interactions appear to be important for the regio-selective process, no details are given on the enantioselective transformation. Also, as the authors point out, the intermediacy of an allenyl-Cu species has already been demonstrated by Buchwald and Hoveyda.

Response: Thanks for the comments and suggestion. We agree that Buchwald and Hoveyda's calculation on propargyl/allenyl copper species offered precedence to their structural information.

However, their reaction was addition to carbonyls and reacted with borane. In our work, we not only provided the experimental evidence for the predominance of allenyl copper by isotope labeling experiments (Fig.3a), but also revealed that the preference for allenyl copper is retained across all the ligands. Our examination of the σ -bond metathesis of allenyl copper with silane is also unprecedented, to our knowledge. Our calculations revealed several key conclusions for further reaction development and asymmetric catalysis in Cu-catalyzed hydrosilylation:

(1) Products are generated via allenyl copper (rather than the less stable propargyl copper) via 4-membered (for allenyl silane) or 6-membered metathesis transition structures (for propargyl silane), respectively.

(2) The 6-membered TSs leading to propargyl silane formation is usually disfavored due to endocyclic steric repulsion. However, with cy-xantphos as ligand, the 4-membered TS is disfavored to such an extent (activation barrier of 14.3 kcal/mol for dppp vs 22.3 kcal/mol for cy-xantphos ligated pathway), while the 6-membered TSs leading to propargyl silane formation is now comparatively more favored. In other words, the regioselectivity control for the formation of propargyl silane relies on the ligand's capability to stabilize the 6-membered TS and/or destabilize the 4-membered TS. We revised the description to emphasize the above observations in the revised manuscript.

In addition, following your suggestion we performed high level ab initio calculations on the mechanism of enantiocontrol in the 1,4-hydrosilylation, and the results are summarized in Fig. 3e. The result of our calculations agrees very well with experimental data and suggests that the enantiocontrol is attributed to less distortions in both reacting fragments, which resulted in an earlier transition state in the formation of the major enantiomer. Our calculations also highlight the flexibility of silane in approaching allenyl Cu(I), which may significantly influence the degree of enantiocontrol (for details see the manuscript). These findings are important for further development in Cu-catalyzed hydrosilylation reactions.

Altogether, I do not think this report merits publication in this journal. I would recommend the authors a more specialized venue, such as *Organic Letters* or the *Journal of Organic Chemistry*.

Reviewer #3 (Remarks to the Author):

This present manuscript by Xu and coworkers describes a Cu-catalyzed regio- and enantioselective 1,2- and 1,4-hydrosilylation of 1,3-enynes. Cu-H is regarded as the reactive species to initiate the reaction. The authors demonstrate their method enables both 1,2- and 1,4-hydrosilylation with excellent levels of regioselectivity by ligand-controlled strategy. More importantly, copper catalyzed enantioselective 1,4-hydrosilylation of 1,3-enynes was realized for the first time, giving the

enantioenriched allenylsilanes in excellent ee values. Unfortunately, asymmetric 1,2-hydrosilylation was not successful, only moderate ee could be observed. In addition to scope, a series of experiments are conducted in order to gain mechanistic insight, including DFT calculation studies. For these studies, the authors proposed allenylcopper is probably the reactive intermediate. Along with manuscript the authors provide a detailed supporting information that is high quality and well done.

Overall, the authors have developed a good catalytic method for regioselective preparation allenylsilanes and propargylsilanes, which will be broad of interest to those in the fields of synthesis and catalysis as well as readers of *Nature Communications*. Consequently, I recommend this communication be accepted for publication with minor revisions.

1) In figure 1a, 1,4-hydrosilylation is missing. In Cui's work shown in figure 1b, Re should be replaced by La. In figure 1c, ee value was not given for asymmetric 1,2-hydrosilylation.

Response: Thanks for your correct, we have changed it in the revised figure 1a. Re has been replaced by La in figure 1b. In addition, the ee values of asymmetric 1,2-hydrosilylation have also been provided in figure 1c.

2) In table 1 and table 3, symbol % was given in columns of yield and ee, which make the table not concise. Symbol % should be only given in the first row.

Response: Thanks for your suggestion. To make these tables more concise, symbol '%' has been removed to the first row in both table 1 and table 3.

3) From line 111 to 119, the description of the performance of L₂ to L₄ should added as well as the results of entry 11 in table 3. Otherwise, entry 11 could be removed from the table.

Response: Thanks for your suggestion. The performance of L₂ to L₄ were described in this revised manuscript and entry 11 has been removed from the table.

4) In line 162, the authors mentioned nitrogen containing ligands were screened. But, there was no N-containing ligands shown in Table S1. The very special silane was used in asymmetric 1,2-hydrosilylation, please explain. If possible, please include more detailed optimization data in SI.

Response: Thanks for this valuable question. Results of several nitrogen containing ligands have been provided in this revised supporting information file. During the optimization of this asymmetric 1,2-hydrosilylation reaction different silanes were tested (see below), and we found that

the presence of ether group on the *ortho* position is beneficial for this reaction. In addition, more detailed optimization data has been provided in this revised supporting information file.

5) To demonstrate the potential application of the products, the authors should give more examples of derivatization.

Response: Thanks for your questions. More efforts have been devoted to the derivatization of these products, and several results have been listed on the revised manuscript. Among them, *para*-methoxy substituted allenylic silane can undergo iodoarylation reaction to construct 2-iodo-3-silylindene in the presence of NIS. The asymmetric iodoarylation reaction was also tried, but only racemic product was obtained. In addition, allenylic silane product can be used as chiral silane source for the allene hydrosilylation reaction. Besides, different Denmark coupling reactions and other reactions modes of allenylic silanes were tried, but no desired product was obtained.

unsuccessful Denmark coupling reactions and other derivatizations

Reviewer #4 (Remarks to the Author):

Zhao, Xu and colleagues describe the development of regio- and enantioselective 1,2- and 1,4-hydrosilylation of 1,3-enynes catalyzed by copper-diphosphine complexes. The regioselectivity is ligand-dependent and uniquely prefers a 1,2-hydrosilylation when Cu-DTBM-SEGPHOS or Cu-cy-xantphos is applied. Overall, this work represents a notable addition to the current 1,4-hydrosilylation of 1,3-enynes as well as silane synthesis in general. Publication of this manuscript

in *Nature Communications* may be supported, while the authors are advised to address the following points.

1) The ees are only moderate for the 1,2-hydrosilylations, mostly around 60%, under the circumstance of a sophisticated silane reagent. The enantioselectivity need to be improved with normal silanes.

Response: Thanks for your suggestion. Different silanes have been tested in the asymmetric 1,2-hydrosilylation reaction, and we found that the choice of silane has obvious impact on the enantioselectivity control. Finally, **2k** was selected as the silane source. Tremendous efforts have been devoted to improve the enantioselectivity of 1,2-hydrosilylation reaction, but no better result was obtained (see revised supporting information for the detail data).

2. The authors conclude that the dispersion effects between ligands and reacting reagents determine the regioselectivity. How other factors, such as π - π interactions and steric hindrance, are excluded from this circumstance?

Response: Thanks for the comments and suggestion. We have revised the discussion to include dispersion, π - π stacking interaction (region A) and steric repulsions (regions B, D, E and F). Dispersion and π - π stacking interactions, albeit slightly different in appearance, are both attractive and shown in green, while steric repulsion interactions, which are repulsive, are shown in red in the IRI analysis (see the color bar at the downright corner). In other words, π - π interactions and steric hindrance are not excluded, but are all considered in the discussion as a plus or minus element to the dispersion interactions. The comments on the weak interactions, although being generally qualitative, are based on quantitative data from distortion/interaction analyses and free energy calculations,

making such comparison meaningful and instrumental. We found that in the regiocontrol scenario, the differences of the features in IRI analyses are distinctive and clear-cut. As suggested by one of the other referees, we also added computations on the enantio-control of the 1,4-hydrosilylation. In that case (Fig. 3D), features of IRI analyses were not very distinctive (not shown); rather, distortion/interaction analyses alone was used in the discussion.

3. Only reactions with RSiH_3 sources are provided. How are the reactions for silanes with two R groups?

Response: Thanks for this valuable question. The results of different silanes such as Ph_2SiH_2 , PhMeSiH_2 and Et_2SiH_2 were provided in the revised manuscript. In the racemic 1,4-hydrosilylation reaction, PhMeSiH_2 showed moderate reactivity under standard conditions. However, Ph_2SiH_2 only delivered the corresponding product in about 15% yield and Et_2SiH_2 was inert under the standard conditions. In the asymmetric 1,4-hydrosilylation reaction, PhMeSiH_2 showed low reactivity due to the low temperature.

4. The result for 1,2-hydrosilylation of ArSiD_3 and but-3-en-1-yn-1-ylbenzene (**1a**) is also required. The deuterium-labeling reaction part should be moved to the mechanistic study section.

Response: Thanks for this question. The reaction of ArSiD_3 with **1a** under 1,2-hydrosilylation was conducted, the desired deuterium product was obtained in 87%. These deuterium-labeling reactions have been moved to the mechanistic study section.

5. For Table 1 & 3, CuTc and Cu(OAc)₂ are metal precursors, but not catalysts. Also, since only CuTc and Cu(OAc)₂ are used in Table 1 & 3, the corresponding columns should be removed to simplify the Tables. The DCE column for Table 3 should also be removed.

Response: Thanks for your suggestion. To simplify the Tables, the corresponding columns have been removed.

6. Please provide the clear chemical structures along with the pictures for Fig 3d.

Response: Thanks for the suggestion. We have added the clear chemical structures for Figure 3d.

7. Other errors: For Fig 1a, three 1,2-hydrosilylations are present. The reference numbering for "... Hoveyda and coworkers' observation,⁷⁷" is not correct.

Response: Thanks for your reminding. We are sorry for this mistake; we have corrected it now.

Reviewers' Comments:

Reviewer #1:

Remarks to the Author:

The authors have fully addressed my comments and I support the publication of this work in Nature Communication.

Reviewer #3:

Remarks to the Author:

The suggestions have been addressed well from my viewpoint. I think the revised manuscript can be accepted for publication in Natural Communication as it is.

Reviewer #4:

Remarks to the Author:

The enantioselective 1,2-hydrosilylation is an important part of the manuscript. The authors screened some commercial chiral ligands as well as some solvents, but the enantioselectivity is not improved. From the results, DTBM-SEGPHOS displayed better yield and ee than normal SEGPHOS. I suggest to examine some bulky C₂-symmetric as well as Josiphos-type ligands (Table S1, entry 2) to see if the ee value could be improved.

Reviewer #4 (Remarks to the Author):

The enantioselective 1,2-hydrosilylation is an important part of the manuscript. The authors screened some commercial chiral ligands as well as some solvents, but the enantioselectivity is not improved. From the results, DTBM-SEGPHOS displayed better yield and ee than normal SEGPHOS. I suggest to examine some bulky C₂-symmetric as well as Josiphos-type ligands (Table S1, entry 2) to see if the ee value could be improved.

Response: Thanks for your suggestion. Several bulky C₂-symmetric chiral ligands and Josiphos-type ligands have been examined, but no better result was obtained.